# Incidence of Myocardial Injury in COVID-19-Infected Patients: A Systematic Review and Meta-Analysis

**DOI:** 10.3390/diseases8040040

**Published:** 2020-10-27

**Authors:** Narut Prasitlumkum, Ronpichai Chokesuwattanaskul, Charat Thongprayoon, Tarun Bathini, Saraschandra Vallabhajosyula, Wisit Cheungpasitporn

**Affiliations:** 1Department of Medicine, University of Riverside, Riverside, CA 92521, USA; 2Faculty of Medicine, King Chulalongkorn Memorial Hospital, Chulalongkorn University, Bangkok 10330, Thailand; 3Division of Cardiac Electrophysiology, University of Michigan Health Care, Ann Arbor, MI 48109, USA; 4Department of Medicine, Mayo Clinic, Rochester, MN 55905, USA; charat.thongprayoon@gmail.com; 5Department of Internal Medicine, University of Arizona, Tucson, AZ 85721, USA; tarunjacobb@gmail.com; 6Department of Cardiovascular Medicine, Emory University, Atlanta, GA 30322, USA; saraschandra.vallabhajosyula@emory.edu

**Keywords:** coronavirus, COVID-19, myocardial injury, meta-analysis, systematic review

## Abstract

**Introduction:** The incidence of acute myocardial injury (AMI) among Coronavirus Disease 19 (COVID-19)-infected patients remain unclear. We aimed to conduct a systematic review and meta-analysis to further explore the incidence AMI in these patients. **Methods:** We comprehensively searched the MEDLINE, EMBASE and Cochrane databases from their inception to August 2020. The included studies were prospective or retrospective cohort studies that reported the event rate of AMI in COVID-19 patients. Data from each study were combined using random-effects to calculate the pooled incidence with 95% confidence intervals. **Results:** We identified twenty-seven studies consisting of 8971 hospitalized COVID-19-infected patients. The study demonstrated that 20.0% (95% CI 16.1–23.8% with substantial heterogeneity (I^2^ = 94.9%)) of hospitalized COVID-19 patients had AMI. In addition, our meta-regression suggested that older age, male and comorbidities were associated with a higher risk of AMI. **Conclusion:** The incidence of COVID-19-related myocardial injury ranges from 16.1–23.8%. Further larger studies are anticipated, as the pandemic is still ongoing.

## 1. Introduction

Coronavirus disease 2019 (COVID-19)-infected patients have shown unique characteristics, with higher infection and mortality rates than prior pandemic-associated respiratory viral infections. However, the incidence and pattern of cardiac involvement for this new emerging respiratory viral infection remains unclear. COVID-19-infected patients have shown unique characteristics, with higher infection and mortality rates than prior pandemic-associated respiratory viral infections. Particularly, myocardial involvement in COVID-19 seems to be common compared to previous coronavirus outbreaks, and is associated with higher morbidity and mortality [1]. As the current pandemic has not yet been resolved, acute myocardial injury (AMI)—which is mostly defined by elevated troponin higher than upper normal limit [2]—is still of importance. The exact incidence is yet to be elucidated, especially from regions other than China. 

Hence, we aimed to conduct a systematic review and meta-analysis to further explore the incidence of myocardial injury among COVID-19-infected patients. 

## 2. Materials and Methods

A systematic literature search of Ovid MEDLINE, EMBASE, and the Cochrane Database of Systematic Reviews (from inception to August 2020) was conducted to identify studies evaluating the incidence/prevalence and clinical significance of myocardial injury among COVID-19-infected patients. 

The systematic literature review was undertaken independently by two investigators (R.C. and N.P.) applying a search approach that incorporated the terms “COVID” or “coronavirus” or “SARS-CoV-2” and “myocardial injury’ or “clinical” (Online Appendix A). No language limitation was applied. 

Eligible studies included cross-sectional, case-control, or cohort studies that assessed the incidence/prevalence and clinical significance of myocardial injury in COVID-19-infected patients. Studies had to provide effect estimates for overall incidence, prevalence, and risk ratios with a 95% confidence interval (CI). Inclusion was not limited by study size. Retrieved articles were reviewed individually for their eligibility by the two investigators noted previously. Analyses were performed using STATA version 14.1. Adjusted point estimates from each study were consolidated using the generic inverse variance approach of DerSimonian and Laird, which designated the weight of each study based on its variance [3]. Meta-regression was also performed to explore risk modifiers.

## 3. Results

The final analysis included 27 observational studies with 8971 hospitalized COVID-19-infected patients (Table 1). Our meta-analysis demonstrated that 20.0% (95% CI 16.1–23.8% with substantial heterogeneity (I^2^ = 94.9%)) of hospitalized COVID-19 patients had a myocardial injury manifested mainly by elevated cardiac troponin I levels (Online Appendix A).

To account for demographic data, we performed meta-regression showing that age, gender, region, and CVD comorbidities. Our analysis suggested older age, male, hypertension, diabetes, coronary heart disease and chronic kidney disease correlated with higher incidence of myocardial injury. (all *p* < 0.01) However, no statistical correlation was found between region, race and incidence of myocardial injury (*p* > 0.1) (Online Appendix A).

## 4. Discussion

Our study highlighted that the incidence of COVID-19 myocardial injury has ranged from 16.1 to 23.8% (Table 2). In comparison with previous epidermic Coronavirus, myocardial injury following COVID-19 seemed to be higher—likely due to underreported incidence of the previous diseases which did not reach pandemic state, suggested by higher mortality rates from those previous diseases limiting their spread. Of note, the diverse incidence of AMI could be explained by differences in demographic data and comorbidities which our study suggested. Despite several studies since the beginning of COVID-19 era, our insight into cardiovascular complications remains limited, warranting further data for better understanding.

Based on our meta-regression, our study also supported that older age, male, and comorbidities—particularly hypertension, diabetes, underlying coronary heart disease and chronic kidney disease—were associated with higher risk of myocardial injury incidence. This suggests that these factors may be casual in cardiac injury process. On the other hand, it was deemed primordial to conclude null impact from races and regions, given the paucity of data from countries other than China. Further studies are encouraged to investigate.

COVID-19 myocardial injury occurs through several mechanisms. One is direct cardiac injury on different parts of the heart by viral entry into the cardiomyocytes. The second is microvascular dysfunction as a consequence of severe inflammatory reaction to the virus. With this cascade come endothelial dysfunction and endothelitis, which further worsen cardiac function [2]. This phenomenon may lead to several manifestations of myocardial injury, from myocarditis and arrhythmia, to Takotsubo cardiomyopathy [35].

Intriguingly, the hypercoagulability state is one of the most unique pathophysiologies proposed in COVID-19, which leads to generalized arterial and venous thrombosis [36]. Owing to the dysregulated immune system, especially in severe infection, several interplays between cytokine storm, platelet hyperactivation and altered microvascular permeability result in abnormal coagulation cascades which promote coronary thrombosis, plaque thrombosis, and even stent thrombosis [37]. Recent studies provided data that support this theory, demonstrating massively elevated Von Willebrand factor, D-Dimer and abnormal procoagulant factors, and even the presence of antiphospholipid antibodies [38,39,40]. 

Another mechanism is indirect involvement through imbalance between metabolic demand and cardiac reserve in patients with preexisting cardiac disease. Based on Choundry et al. [41], we can infer that COVID-19 posed patients at higher risks for acute coronary events. Exaggerated inflammatory responses following the infection can stimulate acute plaque rupture, leading to demand-supply mismatch [42]. As a result, acute myocardial infarction ensues, complicating the patient’s prognosis and clinical course. Nevertheless, data paucity has remained in regard to the true incidence between AMI by microvascular dysfunction and coronary thrombosis among COVID-19 patients, given the difficulty in designing such dedicated studies.

Given the consequent high fatality rate, several clinical trials have been investigated to treat and prevent its progression. Combined antibiotics with Azithromycin and Chloroquine, however, did not improve mortality outcomes but lengthened QT interval, posing significant arrhythmias [43]. For novel therapies such as Remdesevir and IL-6 inhibitors, the data are still limited. Recently, Sheng et al. began a clinical trial using Canakinumab to minimize the risk of myocardial injury, which is currently in the enrolling state [44]. At the moment, we do not have any effective treatments which reduce COVID-19 complications. 

Though informative, our study has certain limitations. First, the statistical heterogeneity is sizable. Thus, meta-regression was performed elucidating the contributions from demographical data and patients’ comorbidities. Moreover, the lack of echocardiographic parameters is cumbersome, further precluding proper variable adjustment. However, the use of echocardiograms in COVID-19-infected patients remains limited due to the disease’s high transmission rate. Second, most studies were from China; thus, real-world incidence may be diverse. Nevertheless, our preliminary analysis suggested no significant difference in myocardial injury incidence. Many more studies from countries other than China are required to demonstrate such diversity. Lastly, true incidence could be overestimated, as most studies used troponin as a marker of cardiac injury, which is not specific to COVID-induced cardiac injury alone, but also to ACS, heart failure, arrhythmia, and so on. Thus, interpretation should be carefully discerned.

## 5. Conclusions

Our study showed the most updated incidence of COVID-19-related myocardial injury, which ranges from 16.1–23.8%. However, as the pandemic has not yet reached the turning point, further studies investigating this relationship with a larger sample size are anticipated.

## Figures and Tables

**Table 1 diseases-08-00040-t001:** Study characteristics.

Study Name	Country	Total Numbers (n)	Mean Age (Years Old)	Sex (Male%)	Hypertension (%)	DM (%)	CHD (%)	CKD (%)	Myocardial Injury Incidence (%)
He [4]	China	54	68	n/a	24.1	14.8	5.3	n/a	44
Huang [5]	China	41	49	73	15	20	15	n/a	12
Wang [6]	China	138	56	54.3	31.2	10.1	14.5	2.9	7
Zhou [7]	China	191	56	62	30	19	8	1	17
Liu [8]	China	56	53.75	55	18	7	3.6	1	13
Chen [9]	China	150	59	44	33	13	6	n/a	20
Shi [10]	China	416	64	49.3	31	14	16	3.4	20
Deng [11]	China	225	54	55	26	12	8	n/a	29
Yang [12]	China	52	59.7	67	n/a	9	5	n/a	23
an [13]	China	135	47	53.3	9.6	8.9	5.2	n/a	7
Cao [14]	China	102	54	52	27.5	10.8	4.9	3.9	15
Guo [15]	China	187	58.5	n/a	32.6	15	11.2	3.2	28
Tao [16]	China	312	69.2	60	57.1	38.8	29.8	3.21	33
Tu [17]	China	174	53.7	45.4	21.2	9.8	9.2	n/a	14
Du [18]	China	179	58	54.2	32.4	18.4	16.2	n/a	23
Xu [19]	China	88	57.1	40.91	23	12.5	7.95	n/a	8
Wu [20]	China	201	51	63.1	19.4	10.9	8	n/a	4
Wei [21]	China	101	49	53.5	21	13.9	5	n/a	16
Ni [22]	China	176	67	57.39	49	26	14	n/a	28
Li [23]	China	548	60	50.9	30.3	15.1	6.2	1.8	22
Yu [24]	China	226	64	61.5	42.5	20.8	9.7	0.35	27
Feng [25]	China	476	53	56.9	23.7	10	8	0.8	11
Lombardi [26]	Italy	614	67	70.8	57	24	22.3	17.9	45
Javanian [27]	Iran	100	60	51	32	27	20	12	14
Saleh [28]	Iran	386	59.5	61.1	36.8	34.5	25.1	4.1	30
Chung [29]	South Korea	110	56.9	43.6	33.6	16.3	9.1	n/a	12
Richardson [30]	USA	3533	63	60.3	56.6	33.8	11.1	8.5	23

**Abbreviations:** CVD: Coronary heart disease; CKD: Chronic kidney disease; DM: Diabetes. n/a: Not applicable.

**Table 2 diseases-08-00040-t002:** Cardiovascular manifestation among recent epidermic/pandemic Coronavirus.

Summary of Cardiovascular Presentations among Outbreak Coronavirus			
Outbreak Period	2003	2015	Current
Comparison	SARS	MERS	COVID-19
Pathophysiology	Exaggerated immune response [31]	Unclear	Cytokine storm, direct viral injury, plaque instability
Myocardial injury incidence	No clear data		Varied from 16.1–23.8%
Cardiovascular manifestation	Subclinical diastolic dysfunction, tachycardia, hypotension, cardiomegaly, atrial fibrillation, myocardial ischemia, elevated troponin [31,32,33]	Acute myocarditis, Acute heart failure [34]	Shock, Acute myocarditis, Acute heart failure, Elevated troponin, Arrhythmia [6,30]

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
