# Peer review of "Incidence of Myocardial Injury in COVID-19-Infected Patients: A Systematic Review and Meta-Analysis"

_diseases, 2020, doi:10.3390/diseases8040040_

Round 1
Reviewer 1 Report
The article provides useful information about myocardial injury in COVID-19 patients. However, it could be acceptable after major revision.
The epidemiological data of the used set is not provided.
The association of myocardial injury with another factors as country, region, age, sex, gender, the accompanying disease conditions must be analysed and discussed.
The provided data, figures and tables are very limited, despite the analysis 19 different papers.
The current clinical trials must be mentioned
A comparison should be provided for myocardial injury in SARS CoV2 and other forms of CoVs, especially HCoV, MERS CoV and SARS CoV.
Author Response
Reviewer 1.
The article provides useful information about myocardial injury in COVID-19 patients. However, it could be acceptable after major revision.
Response: We thank you for reviewing our manuscript and for your critical evaluation.
Comment #1 The epidemiological data of the used set is not provided.
Response: The reviewer raises important point. We agree with the reviewer. We reviewed included studies again and additionally added more epidemiological data in our table 1 as suggested.
Comment #2. The association of myocardial injury with another factors as country, region, age, sex, gender, the accompanying disease conditions must be analyzed and discussed.
Response: We appreciate the reviewer’s input. We agree with this important point. We hence performed meta-regression to sub- analyze the association with all factors described above as well as discussed more in our discussion part in paragraph 2 as suggested.
Comment #3. The provided data, figures and tables are very limited, despite the analysis 19 different papers.
Response: Appreciate reviewers’ suggestion. The reviewer raises an important point. We Since these figures and tables are not high yield. We decided to replace these with figure comparing SARS, MERS and COVID-19 myocarditis for general description.
Comment #4. The current clinical trials must be mentioned
Response: We agree with the reviewer. We have more elaborated recent clinical trials particularly novel treatments for COVID-19, especially Canakinumab which is currently being enrolled by Cleveland Clinic investigators.
Comment #5. A comparison should be provided for myocardial injury in SARS CoV2 and other forms of CoVs, especially HCoV, MERS CoV and SARS CoV.
Response: The reviewer raises an important point. We agree and we have created additional figure comparing myocardial injury among recent epidermic/pandermic Coronavirus as suggested.
We greatly appreciated the editors’ time and comments to improve our manuscript. The manuscript has been improved considerably by the suggested revisions.

Reviewer 2 Report
In the manuscript titled “Incidence of Myocardial Injury in COVID-19 Infected Patients: A Systematic Review and Meta-analysis”, authors conducted meta-analysis and explored the incidence of acute myocardial injury (AMI) among COVID-19-infected patients. 19% patients had AMI in 1,777 hospitalized COVID-19-infected patients from 13 studies. Generally, the authors provide a pioneer investigation on COVID-19-related AMI, which has significant clinical relevance.
One minor concern is observed. Authors claimed “To the best of our knowledge, this is the first study to show the incidence of COVID-19-related myocardial injury.” However, the incidence was reported in Prog Cardiovasc Dis. 2020 Jun 6;S0033-0620(20)30123-7 two months ago.
Author Response
Reviewer 2
Comment #1. In the manuscript titled “Incidence of Myocardial Injury in COVID-19 Infected Patients: A Systematic Review and Meta-analysis”, authors conducted meta-analysis and explored the incidence of acute myocardial injury (AMI) among COVID-19-infected patients. 19% patients had AMI in 1,777 hospitalized COVID-19-infected patients from 13 studies. Generally, the authors provide a pioneer investigation on COVID-19-related AMI, which has significant clinical relevance.
Response: We thank you for reviewing our manuscript and for your critical evaluation. We appreciate reviewer’s suggestion.
Comment #2. One minor concern is observed. Authors claimed “To the best of our knowledge, this is the first study to show the incidence of COVID-19-related myocardial injury.” However, the incidence was reported in Prog Cardiovasc Dis. 2020 Jun 6;S0033-0620(20)30123-7 two months ago.
Response: We appreciate the reviewer’s input. We appreciate reviewer’s suggestion. We have amended the statement above. However, our authors believed this is the most updated studies since the we have included Lombadi et al which was recently published in August 2020. In addition, We also additionally added more epidemiological data in our table 1, and we have created additional figure comparing myocardial injury among recent epidermic/pandemic Coronavirus.
We greatly appreciated the editors’ time and comments to improve our manuscript. The manuscript has been improved considerably by the suggested revisions.

Reviewer 3 Report
This is a brief report based on a literature survey on COVID-19 related acute myocardial injury (AMI). The authors have found only 13 studies out of thousands of literature examined (title and abstract). Only 19 papers the authors have read throughly. The limitations are;
1) Why this review is required? when I searched there are few recent reviews on the same subjects. A couple of examples;
Acute myocardial injury in patients hospitalized with COVID-19 infection: A review. Prog Cardiovasc Dis. 2020 Jun 6:S0033-0620(20)30123-7
Myocardial and Microvascular Injury Due to Coronavirus Disease 2019.
Eur Cardiol. 2020 Jun 23;15:e52. doi: 10.15420/ecr.2020.22. eCollection 2020 Feb.2) I am not sure the quality of the data deserve a publication
3) The authors are not introducing not even the AMI
4) Figure 1 is not adding since the design is not complicated enough to present graphically
Author Response
Reviewer 3
This is a brief report based on a literature survey on COVID-19 related acute myocardial injury (AMI). The authors have found only 13 studies out of thousands of literature examined (title and abstract). Only 19 papers the authors have read throughly.
Response: We thank you for reviewing our manuscript and for your critical evaluation. We appreciate reviewer’s suggestion.
Comment #1. The limitations are; 1) Why this review is required? when I searched there are few recent reviews on the same subjects. A couple of examples;
Acute myocardial injury in patients hospitalized with COVID-19 infection: A review. Bavishi C, Bonow RO, Trivedi V, Abbott JD, Messerli FH, Bhatt DL.Prog Cardiovasc Dis. 2020 Jun 6:S0033-0620(20)30123-7
Myocardial and Microvascular Injury Due to Coronavirus Disease 2019.
Montone RA, Iannaccone G, Meucci MC, Gurgoglione F, Niccoli G.Eur Cardiol. 2020 Jun 23;15:e52. doi: 10.15420/ecr.2020.22. eCollection 2020 Feb.
Response: We appreciate the reviewer’s input. The reviewer raises very important point. There have been previous review articles regarding COVID-19 and acute myocardial injury. Our aim is to provide the most updated information. There are still some gaps which are needed to be filled in such as any factors that portend more risk of AMI which our revised version was discussed. Also, this current version also provided a bit more diverse as we have included studies from USA, Italy and South Korean to provide further information.
Comment #2. I am not sure the quality of the data deserve a publication
Response: We respect the reviewer’s comment and we thus have extensively revised by filling in parts which previous studies have not shown before especially inclusion of studies from Europe and United states as well as perform complex analysis with meta-regression, ultimately to improve our manuscript as suggested. In addition, We also additionally added more epidemiological data in our table 1, and we have created additional figure comparing myocardial injury among recent epidermic/pandemic Coronavirus.
Comment #3. The authors are not introducing not even the AMI
Response: The reviewer raises important point. Appreciate reviewer’s suggestion. We have amended this part in our revised version as suggested.
Comment #4. Figure 1 is not adding since the design is not complicated enough to present graphically
Response: Appreciate reviewer’s suggestion. We agree and thus we decided to show more informational data by substituting our figure 1 with comparison of cardiovascular manifestation in each type of coronavirus instead as suggested.
We greatly appreciated the editors’ time and comments to improve our manuscript. The manuscript has been improved considerably by the suggested revisions.

Reviewer 4 Report
authors performed a review on the incidence of AMI among COVID-19 confirmed patients.
they wrote that their work was performed using two database from 1946 and from 1988 the second one but Covid-19 is a disease found in 2019 so methodology is totally wrong.
furthermore, the articles that they included performed a clinical difference between AMI for thrombosis of large vessels and AMI due to microvascular damages but they did not comment this relevant clinical difference so escaping it also discussing the outcomes of these patients.
in this form I found the manuscript not useful for any type of readers.
Author Response
Reviewer 4
Authors performed a review on the incidence of AMI among COVID-19 confirmed patients.
Response: We thank you for reviewing our manuscript and for your critical evaluation. We appreciate reviewer’s suggestion.
Comment #1. They wrote that their work was performed using two databases from 1946 and from 1988 the second one but Covid-19 is a disease found in 2019 so methodology is totally wrong.
Response: The reviewer raises important point. We agree with the reviewer, we have amended that part by taking out those statements as suggested.
Comment #2. Furthermore, the articles that they included performed a clinical difference between AMI for thrombosis of large vessels and AMI due to microvascular damages but they did not comment this relevant clinical difference so escaping it also discussing the outcomes of these patients.
Response: We appreciate reviewer’s suggestion. In general, there are scarce data reporting the true prevalence COVID-19 associated AMI by either coronary thrombosis or microvascular dysfunction given paucity of data and hard-to-prove causality. We further discussed this crucial element in our discussion part in paragraph 3-4 as suggested.
Comment #3. In this form I found the manuscript not useful for any type of readers.
Response: We respect the reviewer’s comment and we thus have extensively revised and filled in some parts which previous studies have not discussed before especially a figure comparing myocardial injury among each type of coronavirus related cardiovascular dysfunction, meta-regression to analyze the impacts of demographical data and comorbidities as well as update the most recent incident especially adding studies from USA, Italy and South Korea to better provide more diverse information as suggested.
We greatly appreciated the editors’ time and comments to improve our manuscript. The manuscript has been improved considerably by the suggested revisions.

Round 2
Reviewer 1 Report
The manuscript was substantially improved
Minor comments
Improve the legend of table 1.
Figure 1 could be converted to Table and references added to their corresponding locations inside the table.
Author Response
Reviewer 1
The manuscript was substantially improved.
Response: We thank you for reviewing our manuscript and for your critical evaluation.
Minor comments
Comment #1. Improve the legend of table
Response: Appreciate reviewer’s suggestion. We have amended the legend of our table especially for clarity of data presentation, especially units of each element.
Comment #2. Figure 1 could be converted to Table and references added to their corresponding locations inside the table.
Response: Appreciate reviewer’s suggestion. We have converted from figure 1 to table 2 and cited the relevant citations behind the important messages.
We greatly appreciated the reviewer’s time and comments to improve our manuscript. The manuscript has been improved considerably by the suggested revisions.

Reviewer 4 Report
i appreciated authors revision but i actually do not consider suitable the manuscript for publication because basic misunderstandings on articles' selection and methodological selection of type of myocardial injuries.
Author Response
Reviewer 4
Response: We thank you for reviewing our manuscript and for your critical evaluation. Appreciate reviewer’s suggestion. Due to dynamic changes in COVID-19 research and data, perfection in providing the accurate data would be extremely difficult. Despite these inevitable limitations of the study, we believe our project currently provides the most updated data regarding COVID-19 myocardial injury. For methodologies clarity, we have additionally provided our search and selection process in the supplementary file. We would like to apologize the reviewer for this limitation
We greatly appreciated the reviewer’s time and comments to improve our manuscript. The manuscript has been improved considerably by the suggested revisions.
